# Towards Formality-Aware Neural Machine Translation by Leveraging Context Information

**Dohee Kim**[*]    **Yujin Baek**[*]    **Soyoung Yang**    **Jaegul Choo**

KAIST AI

{dohee1121, yujinbaek, sy_yang, jchoo}@kaist.ac.kr

## Abstract

Formality is one of the most important linguistic properties to determine the naturalness of translation. Although a target-side context contains formality-related tokens, the *sparsity* within the context makes it difficult for context-aware neural machine translation (NMT) models to discern them properly. In this paper, we introduce a novel training method to explicitly inform the NMT model by pinpointing key informative tokens using a formality classifier. Given a target context, the formality classifier guides the model to concentrate on the formality-related tokens within the context. Additionally, we modify the standard cross-entropy loss, especially toward the formality-related tokens obtained from the classifier. Experimental results show that our approaches not only improve overall translation quality but also reflect the appropriate formality from the target context.

## 1 Introduction

Translation quality is determined not only by preserving semantics across languages but also by conforming to an appropriate syntax. Formality is one of the important syntactic properties, including honorifics. For example, as shown in Table 1, $S_2$ can be translated into two variants: informal ($T_2^A$) and honorific ($T_2^B$). At the sentence level, the neural machine translation (NMT) models can generate translations conditioned on a provided formality (Niu et al., 2017, 2018; Feely et al., 2019). While this approach sheds some light on handling the stylistic variations in NMT, it is still difficult to control the subtle differences within a class based on pre-defined formality class. To overcome this issue, we focus on the beyond sentence-level NMT setting, where the formality information remains intact in the target-side context. For instance, in Table 1, $T_1^A$ and $T_1^B$ have specific formality tokens highlighted in pink and blue, respectively.

| | Source sentence |
|---|---|
| $S_1$ | Do you have plans this weekend? |
| $S_2$ | I'm actually pretty busy this weekend. |
| | **Target sentence** |
| $T_1^A$ | 이번 주말에 다른 계획 있니 ? |
| $T_2^A$ | 사실 나 이번 주말에 매우 바빠 . |
| $T_1^B$ | 이번 주말에 다른 계획 있으세요 ? |
| $T_2^B$ | 사실 저 이번 주말에 매우 바빠요 . |

Table 1: Example from OpenSubtitles (En→Ko). Translation of the current target sentence ($T_2$) depends on the target-side context ($T_1$). When honorific context ($T_1^B$) is given, it is natural that the subject("저") or ending("바빠요") of the current target sentence is also translated as an honorific ($T_2^B$). A subtle level of formality can be captured in the target-side context.

Since recent context-aware NMT models (Maruf et al., 2019; Zhang et al., 2020; Bao et al., 2021) incorporate the context into the NMT task, these models can be utilized to infer the appropriate formality from the context and, accordingly, generate consistent translations. However, due to the *sparsity of context* (Lupo et al., 2021), it is challenging to capture a few formality-related words scattered within the context. Yin et al. (2021) guides the model to concentrate on the relevant contexts by collecting human-annotated contextual dataset, but this method resorts to manual annotations, which is time-consuming and labor-intensive. Additionally, post-editing is another approach for addressing the delicate formality, which needs cognitive efforts.

Therefore, we propose a new training approach to encourage the NMT model to pinpoint informative context tokens using a formality classifier. We obtain an importance distribution of each context from the classifier, which has more weight on formality-related tokens. We then regularize the attention map of the NMT decoder toward it. Besides this instance-level guidance, we extract salient tokens contributing to the formality class from the

---

[*]Equal Contributions.

classifier. We assign more weights to the loss for these tokens to strengthen formality-controlled generation. Since the NMT model and classifier share vocabulary and tokenizer, the classifier can be flexibly integrated to generate formality-sensitive translations.

Our contributions can be summarized as follows:

- We propose a novel training strategy to extract sparsely distributed formality-related information using a formality classifier.

- We emphasize that our approach is easily adaptable to the NMT model since the classifier and NMT model have the same vocabulary and tokenizer.

- We prove that regularized attention aligns with human perception, and empirical results show that our approach improves translation quality in terms of BLEU and human evaluation.

## 2 Related Work

As NMT models have reached a considerable level, handling of discourse phenomena (Bawden et al., 2018; Müller et al., 2018; Voita et al., 2019b) has become crucial to improve translation quality. Formality is an important linguistic feature addressed by context-aware NMT models for coherent and fluent discourse.

Since a few tokens within a sentence reveal the formality, context-aware NMT models have difficulty to training and evaluating consistent stylistic translation. First, due to this *sparsity* (Lupo et al., 2021) that also occurs within the context, a training alternative is needed to amplify the weak training signal. In addition, BLEU (Papineni et al., 2002), a standard translation quality metric, often fails to capture improvements, which bring subtle differences. To better perceive these differences in translation, human evaluation is sometimes performed in parallel (Voita et al., 2019a; Xiong et al., 2019; Freitag et al., 2022). Our approach directs the model to reach the heart of minor but decisive formality tokens. As a result, our experiments include the performance in terms of BLEU and human evaluation.

## 3 Methods

Given a source document $\mathbf{X} = \{x^1, x^2, ..., x^N\}$ as a sequence of $N$ sentences, context-aware NMT models translate a target document $\mathbf{Y} =$

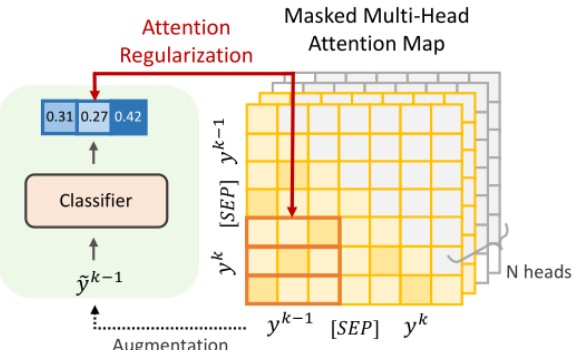

Figure 1: Instance-level context attention regularization mechanism. We obtain a continuous distribution of the target context by making perturbations ($\tilde{y}^{k-1}$) to the target context $y^{k-1}$. We then regularize two attention heads of the top decoder layer.

$\{y^1, y^2, ..., y^N\}$. Basically, the probability of translating $x^k$ into $y^k$ given the context $C$ is defined as:

$$p(y^k|x^k) = \prod_{t=1}^{T} P(y_t^k|x^k, y_{<t}^k, C)$$

where $y_t^k$ represents the $t^{th}$ token of $k^{th}$ sentence and $y_{<t}^k$ represents all the previous target tokens of $k^{th}$ sentence. When the context size of the source and target is one, the context $C$ is $\{x^{k-1}, y^{k-1}\}$, where the target context $y^{k-1}$ is the previously generated target sentence.

However, to generate consistent translation reflecting the sophisticated formality of the target context $y^{k-1}$, training and inference strategies are modified. When training with teacher forcing, the cross-entropy loss is not applied to the target context. During the model inference, the target context is not generated but provided as a condition to avoid exposure bias.

### 3.1 Formality Classifier as Context Guider

We extract formality-involved clues of each target context at the instance-level and formality-sensitive token set within a whole training corpus at the global level (Ramon et al., 2020).

**Formality Classifier** We train a binary classifier to classify whether the sentence is honorific or not. The classifier consists of scaled dot-product attention (Vaswani et al., 2017), linear, and softmax layer consecutively. Since formality-labeled monolingual data is unavailable for Koreans, we automatically label portions of the pre-training corpus by identifying several honorific endings as heuristic rules. We hypothesize that the classifier, which

extrapolates from a few heuristic rules, learns abundant formality-related information. The classifier leverages the same vocabulary and tokenizer as the NMT model, resulting to control of the NMT generation immediately. A detailed description of the training formality classifier can be found in Appendix A.1.

**Instance-level context attention regularization**
To guide the model to distinguish the importance within the target context, we obtain importance distribution for each target context from the classifier through LIME algorithm (Ribeiro et al., 2016). As shown in Figure 1, the algorithm is used to generate token-level distribution based on the impact on predicting formality class by removing random tokens from the target context. Therefore, the distribution has more weight on the formality-involved tokens.

We apply attention regularization to encourage the attention of the NMT model to be aligned with the normalized token-level distribution generated by the classifier. We regularize the self-attention from two heads of the top decoder layer. The loss for attention regularization is :

$$\mathcal{L}_{AR} = -\alpha KL(p_{model-attn} || p_{classifier-attn})$$

where $\alpha$ is a weight scalar to balance between translation loss and attention regularization loss.

**Global-level token weighted loss** To further induce the model to generate appropriate formality tokens, we propose a training method that assigns more weights to a set of formality tokens extracted from the classifier. Inspired by Sun and Lu (2020), the salient formality token set $F$ for each class is obtained based on the attention score, which means global importance within a corpus.

The attention score for an arbitrary token $i$ is:

$$a_i = \frac{\mathbf{h}_i^\top \mathbf{V}}{\lambda}$$

where $\mathbf{h}_i$ is the embedded representation of token $i$, $\mathbf{V}$ is a learnable query vector, and $\lambda$ is a scaling scalar.

To prevent the sparse distribution of formality tokens from being diluted during training, the loss for these tokens is emphasized by applying more weights.

The modified cross-entropy of translating $x^k$ into $y^k$ is as follows:

$$\mathcal{L}_{MT} = -\sum_{t=1}^{T} w_k log p(y_t^k | x^k, y_{<t}^k, C)$$

where $w_k > 1.0$ if $y_t^k$ is a token belonging to $F$, and $w_k = 1$ otherwise.

# 4 Experimental Settings

## 4.1 Data

We conduct experiments for two language pairs: English→Korean/Japanese.

**En→Ko** We pre-train sentence-level baseline on a large parallel corpus of 1.5M sentence pairs in AI-HUB [1]. We use two fine-tuning datasets: OpenSubtitles2018 (Lison et al., 2018) and AI-HUB dialogue corpus. For OpenSubtitles dataset, noisy parallel sentences are filtered out through LABSE (Feng et al., 2020) for sentence alignment. We split the refined dataset into 63K for training, 3.3K for validation, and 2.6K for testing.

**En→Ja** We use a publicly available pre-trained model trained on JParaCrawl (Morishita et al., 2020) of 22M sentence pairs. The Business Scene Dialogue (BSD) (Rikters et al., 2019), AMI Meeting (Rikters et al., 2020) and OpenSubtitles2018 (Lison et al., 2018) datasets are combined for fine-tuning. The dataset consists of 97K training data, 63K validation data, and 67K test data.

## 4.2 Models

We consider three models as baselines; *Sent-level* is an abbreviation for the sentence-level model without any context. *Concat-level* has both previous source and target sentences as context, but for the previous target sentence, its loss is not applied during the training phase. For the inference, the referential target context sentence is provided as prefix tokens. *Tag context* model is given a formality tag of the classifier as a target-side context. *Concat-level + Tag* model is given both a formality tag and previous target sentence as a target-side context.

In experiments, we focus on the single-encoder architecture that concatenates the context and the current sentences (Bao et al., 2021); based on the findings conducted by Lopes et al. (2020) that this simple approach demonstrates competitiveness or even surpasses its more complex counterparts.

## 4.3 Results

Table 2 shows the performance of our approaches and baselines. We achieve moderate improvements

---

[1] www.aihub.or.kr (Data usage permission is required.)

|  | En→Ko | | En→Ja |
|---|---|---|---|
|  | **AI-HUB** | **OpenSub** | **AMI+BSD +OpenSub** |
| Sent-level | 26.15 | 12.38 | - |
| Concat-level | 27.64 | 20.14 | 12.77 |
| Tag context | 26.98 | 19.46 | 12.07 |
| Concat-level + Tag | 27.82 | 19.93 | 13.47 |
| Ours (*WL*) | 27.85 | 20.44 | 13.53 |
| Ours (*AttnReg*) | 27.86 | 20.38 | 12.91 |
| Ours (*WL+AttnReg*) | **27.90** | **20.70** | **13.67** |

Table 2: Experimental results reported in BLEU. *AttnReg* denotes model with instance-level context attention regularization. *WL* indicates model with global-level token weighted loss. For AI-HUB and AMI+BSD+OpenSub datasets, the improvement of our model (*WL+AttnReg*) is statistically significant at p < 0.05 compared to the *Concat-level* baseline. For Open-Sub dataset, it is statistically significant at p < 0.1.

for each of our models in BLEU (Post, 2018) [2], even though the *Concat-level* is known as a dominant baseline (Bao et al., 2021), where the target context itself is provided as a condition. When both instance-level context attention regularization and global-level token weighted loss are combined, the BLEU score was the highest.

We also conduct a human evaluation on each of 50 examples from the *Concat-level* baseline and our final model (*WL+AttnReg*) where both methods are applied. Following Rao and Tetreault (2018); Niu and Carpuat (2020), we adopt two evaluation criteria: formality consistency and meaning/content preservation.

**Formality consistency** We asked 11 human participants the question, "Choose the sentence that reflects the formality of the previous contextual sentence.". Our model acquires 93% of the total votes, meaning that our model better reflects the subtle stylistic differences. The kappa coefficient, which indicates agreement between 11 annotators, is 0.70.

**Meaning/content preservation** We asked the other 9 human participants the question,"Given the reference translation, which one of the two generated translations, regardless of formality, better preserves its meaning?". An option "It's hard to say which better preserves the reference meaning." was added. As a result, 62% of examples were considered both are comparable, 28% were judged as better translated by our model.

---

[2]We measure SacreBLEU scores with the signature BLEU |nrefs:1 |case:mixed |eff:no |tok:{ko,ja}-mecab|smooth:exp |version:2.2.0.

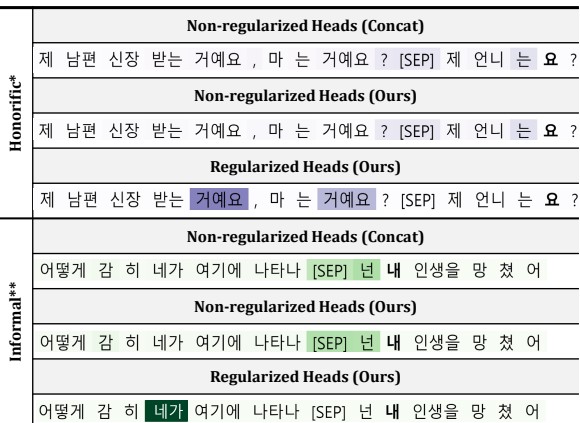

* Is my husband receiving an organ transplant or not? And what about my older sister?
** How dare you show up here? You ruined my life.

Figure 2: Qualitative examples are visualized based on attention weights. Higher intensity of the color indicates higher attention weight. The honorific example is highlighted in purple, while the informal example is highlighted in green.

## 5  Analysis

**Attention Visualization** Figure 2 illustrates how attention weights from regularized and non-regularized heads are distributed when generating formality-related tokens, which is marked in bold. Regularized heads (ours) in each example are visualized with the averaged attention weights over two regularized heads in *AttnReg*. Non-regularized heads (concat) and non-regularized heads (ours) are highlighted using the averaged attention weights over all non-regular heads in concat-level and *AttnReg*, respectively. We observe that regularized heads pay proper attention to formality-related tokens within the target context, such as endings ("예요") or nominative markers ("가"); on the other hand, non-regularized heads seem to focus mostly on the previous few tokens. This experiment shows that the model is able to learn various end expressions via the classifier, and the classifier can control the formality beyond rule-based learning.

**Alignment between Regularized Heads and Human Perception** Attention regularization makes our model explicitly refer to key syntactic tokens that determine the formality of contextual sentences. We investigate how the attention weights of regularized heads are aligned with human perception. The attention weights of regularized heads in our model are shown to be adequately aligned with human perception showing the highest similar-

ity score of 0.669 [3], while non-regularized heads hardly align with human perception: 0.033 (non-regularized heads for our model) and 0.029 (non-regularized heads for concat-level baseline).

## 6 Conclusion

To generate consistent translations reflecting the formality of the target-side context, we propose a new training approach to focus on formality-involved tokens by leveraging the formality classifier. Our approach results in convincing improvements in translation quality as measured by BLEU and human evaluation. As a result, we suggest a lesson that the formality classifier is easily applicable to NMT model as a guider.

## Limitations

We aim to generate appropriate translations in terms of syntactic formality. The improvements achieved are likely to be subtle; however, these subtle differences are not properly captured by BLEU. To better identify these differences, we perform human evaluation, which incurs an additional expense. Although we appreciate the importance of introducing an automatic metric to achieve scalability, we focus more on generating formality-related changes in translation for now, leaving it to future work.

## Ethical Consideration

Since our work is in the field of controlled text generation, if an undesirable formality of target-side context is given, our model is amenable to leveraging it. However, we propose our work in anticipation of positive applications such as helping business conversation or academic writing. We also make sure that our academic use of AI-HUB dataset is consistent with the intended use demonstrated in its license.

## Acknowledgement

This work was supported by SAMSUNG Research, Samsung Electronics Co., Ltd., Institute of Information & communications Technology Planning & Evaluation (IITP) grant funded by the Korea government (MSIT) (No.2019-0-00075, Artificial Intelligence Graduate School Program (KAIST)), and the National Research Foundation of Korea (NRF) grant funded by the Korea government (MSIT) (No. NRF-2022R1A2B5B02001913).

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

# A   Appendix

## A.1   Training Formality Classifier

For English-Korean language pair, we automatically label monolingual pre-training corpus by identifying some pre-defined honorific endings as heuristic rules ("요", "시다", "니다", "시죠", "습니까", "입니까"). The Korean corpus is randomly sampled and split into 6.4K for training, 1.6K for validation, and 1.6K for testing.

For English-Japanese language pair, we use Nădejde et al. (2022) dataset to train the binary formality classifier. The dataset consists of 2K training data, 0.6K validation data and 0.6K test data, which comes from two domain: Telephony and Topical-Chat (Gopalakrishnan et al., 2019). To use the same tokenizer and vocabulary as the NMT model, we use the sentence-piece model and vocabulary provided by Morishita et al. (2020), which are used for pre-training. The accuracies of Korean and Japanese classifiers are 98.7% and 94.8%, respectively.

## A.2   LIME algorithm

We conduct experiments varing the number of perturbations ($\tilde{y}^{k-1}$ in Figure 1): we choose 100 after experimenting with 20, 50, 100, 500 and 1000.

## A.3   Global-level token loss weight

The hyperparameter $w_k$ was tuned in the range [1.1, 1.25, 1.4, 1.5, 1.7, 1.75, 2.0, 2.5, 3.0].

## A.4   Analysis of Global-level token weighted loss

To demonstrate the effectiveness of the global-level token weighted loss (*WL*) qualitatively, we examine how confidently our model (*WL*) generates appropriate formality token compared to *concat-level* baseline. As shown in Table 3, our model succeeds to predict formality-related token "요" accurately. Since the target-side context ($T_1$) represents honorific context colored as blue, the current target sentence should maintain an honorific tone. However, after "뭐라고", the *concat-level* model ($T_2^{\text{concat}}$) yields "?" with 66.1% in place of "요" (29.5%), which completes an informal sentence. However, our model ($T_2^{\text{Ours}}$) produces an honorific token "요" with 58.6%, instead of generating "?" with 38.3%. It implies that *WL* helps the NMT model to generate formality-related tokens.

## A.5   Extracting global-level formality token set

The attention score, which is the standard for extracting honorific tokens, is more than 0.1 for Open-Subtitles dataset and more than 0.2 for AI-HUB dialog and combined Japanese dataset, respectively. Conversely, the attention score is less than -0.2 and -0.1 to extract the informal tokens.

## A.6   Implementation Details

We conduct all our experiments on *fairseq* framework (Ott et al., 2019). For English→Korean language pair, we use 32K shared vocabulary. For English→Japanese language pair, vocabulary is provided with sentencepiece models (Kudo and Richardson, 2018). All context-aware NMT models are an encoder-decoder Transformer architecture (Vaswani et al., 2017), which is concretely *transformer base* (60,528,128 training parameters). We train the models with the optimization details described in (Vaswani et al., 2017). We clip the norm of the gradient to be no more than 0.1, since gradient overflow is detected without clipping.

For model selection, we save the model checkpoint per 1 epoch, and the model with the validation loss in a single run is selected as our final model. It takes about 6 hours on a single NVIDIA A100 machine to train our model with AI-HUB dialog dataset.

## A.7   Acquisition of Human Perception

We derive *human perception* on informative formality-related tokens from annotations made by 13 native Koreans. We ask them to annotate 50 sampled sentences. Given a tokenized sentence in Korean, annotators are asked to select tokens that determine its formality. Majority-voted tokens in the sentence are considered as ground-truth labels. Then, each ground-truth token is assigned an importance score in proportion to the votes cast. We obtain the similarity between this human perception and the attention weights of the models by calculating their dot product.

| | Source sentence |
|---|---|
| $S_1$ | There's an older daughter, as well. |
| $S_2$ | What? |
| | **Target sentence** |
| $T_1$ | 그 아이 누나가 하나 더 있던데 요 . |
| $T_2^{\text{concat}}$ | 뭐라고 ? |
| $T_2^{\text{ours}}$ | 뭐라고 요 ? |

Table 3: Example from OpenSubtitles (En→Ko). Translation of the current target sentence ($T_2$) would depend on the target-side context ($T_1$).